# Applications of 2D Nanomaterials in Neural Interface

**DOI:** 10.3390/ijms25168615

**Published:** 2024-08-07

**Authors:** Shuchun Gou, Siyi Yang, Yuhang Cheng, Shu Yang, Hongli Liu, Peixuan Li, Zhanhong Du

**Affiliations:** 1The Brain Cognition and Brain Disease Institute (BCBDI), Shenzhen Institutes of Advanced Technology, Chinese Academy of Sciences, Shenzhen 518055, China; e1011006@u.nus.edu (S.G.); ysy18987870647@163.com (S.Y.); yh.cheng@siat.ac.cn (Y.C.); s.yang1@siat.ac.cn (S.Y.); pl122@ic.ac.uk (P.L.); 2Guangdong Provincial Key Laboratory of Brain Connectome and Behavior, Shenzhen Institute of Advanced Technology, Chinese Academy of Sciences, Shenzhen 518055, China; 3CAS Key Laboratory of Brain Connectome and Manipulation, Shenzhen-Hong Kong Institute of Brain Science, Shenzhen Institute of Advanced Technology, Chinese Academy of Sciences, Shenzhen 518055, China; 4Shenzhen Fundamental Research Institutions, Shenzhen 518055, China; 5Guangzhou Dublin International College of Life Sciences and Technology, South China Agricultural University, Guangzhou 510642, China; hongli.liu@ucdconnect.ie

**Keywords:** 2D nanomaterials, neural interface, graphene, MXene, biotechnology

## Abstract

Neural interfaces are crucial conduits between neural tissues and external devices, enabling the recording and modulation of neural activity. However, with increasing demand, simple neural interfaces are no longer adequate to meet the requirements for precision, functionality, and safety. There are three main challenges in fabricating advanced neural interfaces: sensitivity, heat management, and biocompatibility. The electrical, chemical, and optical properties of 2D nanomaterials enhance the sensitivity of various types of neural interfaces, while the newly developed interfaces do not exhibit adverse reactions in terms of heat management and biocompatibility. Additionally, 2D nanomaterials can further improve the functionality of these interfaces, including magnetic resonance imaging (MRI) compatibility, stretchability, and drug delivery. In this review, we examine the recent applications of 2D nanomaterials in neural interfaces, focusing on their contributions to enhancing performance and functionality. Finally, we summarize the advantages and disadvantages of these nanomaterials, analyze the importance of biocompatibility testing for 2D nanomaterials, and propose that improving and developing composite material structures to enhance interface performance will continue to lead the forefront of this field.

## 1. Introduction

Neural interfaces are crucial conduits between biological tissues and external de-vices, enabling the recording and modulation of neural activity through electrical, chemical, and optical methods. They are essential in treating neurological disorders, modulating neural circuits, and regulating neural functions [1]. The ideal neural interface should have several key capabilities: the ability to selectively record or stimulate specific neurons and neural tissue, the ability to capture high-resolution neural signals, and stability [2]. Since the pioneering work on neural stimulation through direct electrical current application in the late 19th century, significant research efforts have been dedicated to developing neural interfaces aimed at achieving precise and large-scale recording and modulation of neural activity [3,4,5]. With these developments, neural interfaces have made significant advancements in the mechanistic research and treatment of neurological disorders, such as epilepsy [6], depression [7], Alzheimer’s Disease [8,9], and Parkinson’s Disease [10]. However, with the increasing demands for accuracy in clinical medicine and life science research, simple neural interfaces are no longer adequate to meet the requirements in precision, functionality, and safety [11,12,13].

Nanomaterials are evolving, and their properties are becoming increasingly suitable for applications in neural interfaces [14,15]. Two-dimensional nanomaterials, with their high aspect ratios, high conductivity, and quantum size effects, offer advantages such as improved signal-to-noise ratio and increased charge injection density [16,17,18,19]. Furthermore, the unique physicochemical properties of 2D nanomaterials can enhance functionality, including MRI compatibility, stretchability, and drug delivery [20,21,22,23]. Some 2D nanomaterials also demonstrate biocompatibility with neural tissue and mitigate undesirable tissue reactions such as inflammation and neural cell injury [24,25,26,27]. In recent years, there has been rapid progress in the design of advanced neural interfaces incorporating functional nanomaterials [28,29,30].

In this review, we initially explore the challenges confronting neural interfaces, including issues related to sensitivity, heat, and biocompatibility, as shown in Figure 1. Then we focus on the application of 2D nanomaterials, such as graphene, MXenes, and hexagonal boron nitride, in neural interfaces, with an emphasis on their impact on neural interface performance. Finally, we discuss the challenges and future research directions in this research area, providing some advice for future research.

## 2. Challenges of Neural Interface

Establishing high-quality conduits between neural tissues and devices is a challenging task [31]. In addition to complex functional requirements, there are still many challenges in meeting the needs of advanced neural interfaces, including sensitivity, heat, and biocompatibility. Understanding the causes of these challenges and identifying critical performance metrics is essential.

### 2.1. Sensitivity

Sensitivity affects the precision of the interaction between the interface and neural tissues. The evaluation criteria for sensitivity vary depending on the type of neural interface.

For chemical neural interfaces such as drug-delivery platforms and neurotransmitter sensors [23,32,33,34], sensitivity primarily depends on their ability to respond to target molecules or ions in specific environments [35]. The sensitivity is influenced by factors including the material’s reaction kinetics, environmental responsiveness, and interface interactions. Due to the specificity of different chemical interfaces, the criteria for assessing sensitivity often vary between interfaces. This variability makes it hard to conduct direct comparisons using a single, unified set of parameters.

For purely optical interfaces, optical loss within the optical path poses a significant challenge to sensitivity. Reducing optical loss is crucial as it maximizes the retention of optical signal strength, thereby improving sensitivity [36]. Highly transparent materials, which exhibit low absorption and scattering rates, can minimize the scattering and attenuation of optical signals. This preservation of signal clarity and strength allows for localized transmission, enhancing sensitivity. Moreover, different types and morphologies of neural tissue may require different wavelengths to match the optical penetration depth with the target excitable tissue. Consequently, it is essential to analyze optical transparency in relation to the specific type of light used. With increasing diversity in applications, multifunctional optoelectronic interfaces are employed to meet increasingly complex demands. For these interfaces, sensitivity cannot be simply judged based on transparency. For instance, in artificial retinas, the optical interface materials may require high absorption rates to ensure the sensitivity of photonic signal changes within the interface [37]. Additionally, photo-induced artifacts resulting from the photoelectric effect may also affect sensitivity [38].

The sensitivity of electrical neural interfaces, unlike that of chemical and optical interfaces, is primarily influenced by the interactions occurring at the electrode–tissue interface [39,40,41]. Parameters derived from these interface reactions serve as crucial indicators for optimizing and evaluating the performance of these interfaces [42,43]. These reactions, which facilitate the transition from electronic to ionic flow, can be categorized into two types: capacitive and Faradaic [44]. In capacitive processes, the electrode surface stores charges. A high value of charge-storage capacity (CSC), particularly the cathodic charge-storage capacity (CSCc), indicates a high charge-transfer rate, resulting in increased sensitivity for signal detection. The higher the CSCc, the more efficient the electrode is at storing and transferring charges, which enhances its ability to detect subtle neural signals with greater precision. In Faradaic reactions, electrons cross the electrode–tissue interface, and the sensitivity of neural electrodes to small currents is directly related to the impedance of the electrodes. Lower impedance allows for better detection of small currents, improving the overall sensitivity of the neural interface [13,45].

Additionally, for chemical and optical interfaces, including circuit structures, CSC and impedance may also influence their sensitivity.

### 2.2. Heat

Temperature changes can alter the diffusion rates of substances and directly impact the kinetics of membrane-protein gating. Specifically, the opening and closing rates of transmembrane proteins directly influence the generation and transmission of action potentials, thereby interfering with the responsiveness of the nervous system [46]. A fluctuation in tissue temperature of as little as 1 °C can lead to transient inhibition or the excitation of neurons, affecting the fidelity of neural recordings. Conversely, temperature fluctuations exceeding 4 °C can cause damage to the nervous system [47].

In optical interfaces, heat is a significant limitation of optical stimulation. The absorption of photons by the optoelectronic material generates excited states, the decay of which produces localized heat. Thus, insufficient light-power density or exposure time can result in overheating [48]. Additionally, if the injected heat exceeds the thermal diffusion rate in the target tissue, stimulation parameters considered safe at lower repetition rates may induce damage due to heat accumulation [49]. The thermal effects of optical interfaces can be evaluated based on the material’s light absorption, thermal conductivity, and specific heat capacity [50]. It is noteworthy that within a safe range, photothermal effects can also serve as a kind of stimulation [50,51]. To prevent the overheating of the nervous system during photostimulation, some studies have incorporated temperature sensors within the photostimulation arrays. Rajalingham et al. integrated a thermistor as a temperature sensor in a photostimulation array designed for long-term implantation in the primate cortex [52]. Both sensor measurements and simulations confirmed that the temperature increase under specified photostimulation conditions was less than 0.375 °C. Additionally, a temperature feedback mechanism was incorporated into the system to prevent thermal issues. In addition to incorporating temperature sensors, some studies have used modeling and simulation to design stimulation intensities and methods that do not cause tissue damage [53].

In electrical interfaces, heat issues commonly arise due to resistive heating during the transmission of electrical signals through the interfaces. In the processes of electrode stimulation and signal recording, the neural electrode interfaces with brain tissue, forming a closed electrical circuit. The properties of the electrode material, including conductor resistance, cross-sectional area, length, and conductivity, primarily determine the ability of the neural electrode interface to transport charge. When the charge moves, Ohmic or resistive heating occurs at the neural electrode interface, generating thermal power [54]. Therefore, a high-quality electrical neural interface necessitates an electrode interface with low impedance. Furthermore, to mitigate heating effects, the peak amplitude of electrical stimulation delivered by electrodes is typically kept below 1 V, and a relatively high-duty cycle is employed. Another source of overheating with implanted neural electrodes arises from the heating of electrode tips during MRI scans. To mitigate this issue, electrodes are commonly fabricated from materials with lower paramagnetic properties [55].

Purely chemical interfaces, such as neural drug delivery platforms, typically do not present the challenge of heating. However, for interfaces that include circuit structures, such as neurotransmitter-detection systems, heat issues are similar to those in electrical interfaces.

### 2.3. Biocompatibility

In the design and manufacturing process of neural interfaces, careful attention must be devoted to the biological response to material–tissue contact, especially in the case of invasive neural interfaces [56,57,58,59]. Electrodes with low biocompatibility can trigger inflammatory immune responses upon implantation and lead to the formation of neuroglial scars with surrounding tissues. These reactions degrade the quality of neural signal recordings, and the associated neuronal degeneration may cause irreversible damage to the brain. In the case of the brain, for example, after an interface is implanted, the body’s immune system may recognize it as foreign and initiate an immune response that is generally divided into two overlapping phases: an acute tissue response and a chronic inflammatory response [60]. Mechanical damage and vascular rupture during implantation trigger an acute response immediately after the procedure. This process involves the adsorption of plasma proteins, the activation of the coagulation cascade, and the stimulation of macrophages and microglial cells [61]. Compared to acute inflammation, chronic inflammation caused by minor electrode movements and the inherent immunogenicity of the materials has a more significant long-term impact on signal quality. Mechanical mismatches between the electrode and brain tissue exacerbate the damage to neurons and blood vessels due to micromotion. The activation of astrocytes and microglia results in their accumulation around the electrode, forming scars that isolate neurons from the electrode, thereby impairing signal transmission and reducing recording quality. This leads to the attenuation of electrophysiological signals, increased impedance, and decreased signal-to-noise ratio. It may also affect the electrophysiological properties of the neural tissue and potentially impact neural function over time. The thickness of the scarring can sometimes reach several hundred microns, severely affecting the capture of single-neuron spike signals [62]. In addition, the existence of complexing agents such as chloride ions and reactive oxygen species (ROS) in neural tissues, as well as inflammation-induced hydrogen peroxide (H2O2), can lead to corrosion and affect the performance of chronically implanted interfaces. Additionally, irreversible reactions on the electrode surface of interfaces with stimulation can accelerate corrosion [13].

Aligned with the ISO 10993 series of standards and the ASTM F2129 standard for evaluating medical devices, the following methods in Table 1 are applicable for assessing the biocompatibility of neural interfaces [63,64,65].

Furthermore, for invasive neural interfaces, mechanical strength is also an important indicator of biocompatibility, which is usually quantified by Young’s modulus. The compliance of central nervous system tissues falls within the range of 100 Pa to 10 kPa, as shown in Figure 2. The mismatch in mechanical properties may lead to compression, damage, or irritation of surrounding tissues. Implants can induce acute or chronic inflammation of the tissue, gliomas, and disruption of the blood–brain barrier, which can reciprocally affect the implant, leading to structural degradation, material corrosion, insulation failure, and fluctuations in electrode impedance [66,67,68,69,70].

## 3. Applications of 2D Nanomaterials in Neural Interface

### 3.1. Graphene and Graphene-Based Materials (GBMs)

Graphene is a 2D material formed by carbon atoms with a specialized atomic structure. It consists of a honeycomb-shaped monolayer of carbon atoms, in which the carbon atom forms in-plane σ-bonds with its three neighboring carbon atoms through sp2 (2 s, 2 px and 2 py) hybridized orbitals, while out-of-plane pi-bonds perpendicular to the plane through 2 pz orbitals. The short length of its interatomic covalent σ-bonds (1.42 A) makes its carbon–carbon bonding strength even stronger than that of SP3-hybridized diamonds, granting it extremely excellent mechanical properties. The presence of half-filled pi bands allows for free electron movement, resulting in zero-bandgap conduction and valence bands, which confer excellent electrical and thermal conductivity. Despite its remarkable strength, graphene structural flexibility reduces localized stresses in cells and mitigates tissue damage caused by mechanical mismatches [71]. The low light absorption of graphene, approximately 2.3%, makes it an ideal material for fabricating transparent interfaces [72,73], which is particularly advantageous for optical interfaces like optogenetic stimulation. The magnetization of graphene closely matches that of biological tissues, making it compatible with magnetic resonance imaging (MRI) [38]. Additionally, graphene-based materials (GBMs), such as graphene oxide (GO), reduced graphene oxide (rGO), and graphene nanocomposites, are also strong candidates for the fabrication of neural interfaces [74]. Modified GBMs address the intrinsic hydrophobicity of pristine graphene, facilitating a wider array of fabrication processes and offering diverse properties. Therefore, devices fabricated with GO often need to be reduced to rGO to restore conductivity. The biocompatibility of both graphene and GBMs has been extensively studied [75,76,77,78]. Due to the outstanding properties of graphene and GBMs, they are widely researched as alternatives to metal in the fabrication of neural interfaces. These materials enhance performance in terms of biocompatibility, mechanical strength, transparency, and MRI compatibility.

Fabricating high-performance electrical interfaces using graphene-based materials and testing their biocompatibility is a common research approach. Xiong et al. developed a hydrogel film with high electrical performance by applying a special vacuum filtration process to a chemically converted graphene dispersion exhibiting unusual gel behavior. This hydrogel film achieved a CSC of approximately 160 mC cm⁻² and a CIC ranging from 1 to 6 mC cm⁻², as shown in Figure 3A [71]. In addition, the viscoelastic multilayered graphene hydrogel membrane shows little inflammatory response after an 8-week implantation in rat sciatic nerves. Dong et al. prepared rGO-2D films through a casting reduction method [79]. The rGO electrodes fabricated from these films were able to record field excitatory postsynaptic potentials (fEPSPs) in hippocampal brain slice and acute in vivo local field potentials (LFPs), with a signal-to-noise ratio (SNR) comparable to that of traditional Pt ball control electrodes. The removal of basal plane functional groups and the restoration of the sp²-conjugated graphene network during the reduction process are believed to be the reasons for the improved electrical performance of the samples. The remaining functional groups on rGO are thought to contribute to the material’s pseudocapacitive properties. Modified electrodes promote neuronal functional development and exhibit excellent recording capabilities and biocompatibility. Lim et al. developed an advanced flexible hybrid graphene multichannel array through a multi-step deposition process [80]. The well-stacked graphene–gold composite layers significantly improved charge transfer efficiency and reduced low-frequency impedance. This enhancement resulted in a higher SNR during recording and increased stimulation efficiency of the array. Additionally, immunohistochemical results of neurons and glial cells indicated that the composite layers possess long-term biocompatibility, as shown in Figure 3B.

Based on the 2D structure of nanosheets, the interfaces exhibit excellent mechanical properties. Gao et al. utilized low-temperature chemical vapor deposition to incorporate graphene into the voids of a carbon nanotube network film. This hybrid structure increased the contact area, reduced impedance, and improved the electrical coupling between the film material and neurons while maintaining excellent mechanical properties [20]. The fabricated neural microelectrodes exhibit high flexibility and stretchability, maintaining structural and electrode integrity even under bending, folding, and deformation, as shown in Figure 3C. Results indicate that the samples can form a seamless interface with neural tissue, facilitating long-term neural activity recording, as validated by in vivo experiments with mice. Wang et al. utilized the liquid crystal properties of graphene oxide nanosheets to fabricate fiber electrodes by a wet spinning process. The liquid crystal state of the nanosheets formed a completely ordered and porous structure, facilitating the diffusion of electrolytes and ions into the resulting electrode [81]. As a result, the fiber electrodes exhibited excellent electrical performance, as shown in Figure 3D. After platinum coating modification, the fiber electrodes demonstrated a high CIC exceeding 10 mC cm⁻² and a high SNR of 9.2 dB. Additionally, the stability of the liquid crystal nanosheet structure was validated through fatigue testing and 1000 CV tests.

The magnetization and optical properties of graphene further expand the functionality of neural interfaces. Zhao et al. obtained graphene fiber electrodes with high CIC and minimal to no MRI artifacts at 9.4 T by baking a GO aqueous dispersion in glass tubes, as shown in Figure 3E [22]. The dense stacking of GO sheets endowed the fiber electrodes with beneficial mechanical strength and robustness, which is sufficient to support insertion into the brain. In electrochemical tests, the electrodes demonstrated high CSC (889.8 ± 158.0 mC cm⁻^2^), high CIC (10.1 ± 2.25 mC cm⁻^2^), and high stability (with impedance remaining nearly unchanged over 24 days). Additionally, the capability of the graphene fiber microelectrodes for efficacious deep brain stimulation was demonstrated in a hemi-Parkinsonian rat model. Bakhshaee Babaroud et al. fabricated multilayer graphene electrodes through a wafer-scale transfer-free process and chemical-vapor deposition (CVD) [38]. The electrodes exhibited comparable impedance to gold and platinum electrodes of the same size and shape at 1 kHz. In addition, the electrodes were free of light-sensing artifacts at under 10 Hz of light-pulse irradiation, as shown in Figure 3F, and free of any artifacts under 3T MRI, allowing for multimodal electrical and optical recording and stimulation, as well as compatibility with MRI for neurological studies.

### 3.2. Black Phosphorus (BP)

Since the first discovery of BP in 2014 [82], it has garnered significant attention as a prominent member of 2D semiconductor materials due to its unique geometry and electronic structure. In monolayer BP, each phosphorus atom forms bonds with three adjacent phosphorus atoms through sp3 hybridization, resulting in a puckered structure along the armchair direction and a bilayer configuration along the zigzag direction [83,84], as shown in Figure 4A. Adjacent BP layers are stacked on top of each other by weak van der Waals forces, and this laminar structure ensures the feasibility of stripping single or multiple BP layers from bulk crystals [85]. Unlike graphene, the layers of BP exhibit a folded shape. The anisotropy of the BP crystal structure leads to specific in-plane anisotropy of its optical and electrical properties [86]. Black phosphorus (BP) offers a wider tunable bandgap than many other 2D nanomaterials, ranging from 0.3 eV for bulk BP to 2.0 eV for monolayer BP, enabling absorption across the entire ultraviolet and infrared spectra [87]. BP also degrades into biocompatible phosphates, ensuring its low toxicity in vivo [88]. Furthermore, the photothermal effect of BP has been demonstrated to increase the permeability of the blood–brain barrier, allowing for targeted drug delivery to the brain. Based on unique properties, BP is widely utilized in chemical and optical neural interfaces.

BP-based NIR neuromodulation contributes to the development of nonimplantable optical techniques. Yang et al. proposed a new method for the treatment of epilepsy using biodegradable BP flakes, as shown in Figure 4B [89]. By applying a 980 nm near-infrared laser pulse (4 mW, 15 ms) to neurons with BP flakes, a depolarizing current of 38.13 ± 0.7056 pA was induced on average, which could trigger intercellular calcium dynamics with high spatiotemporal resolution. Tang et al. report an injectable strategy based on black phosphorus nanosheets that successfully performs NIR-driven neurostimulation in cultured cells or animal models and describes controlled modulation of brain functions that control animal behavior, as shown in Figure 4C [90]. This new method is expected to provide an alternative to other neurostimulation interfaces.

In addition, BP can change the adhesion behavior of the interface. Zhang et al. presented a composite patch integrated with BP nanosheets, specifically designed for wet adhesion, as shown in Figure 4D [91]. It has the potential to act as a tissue binder and can rapidly stop bleeding in vitro and in vivo. In the electromyographic test, the patch and Cu foils presented a similar level of sensing capability. Moreover, the biocompatibility was validated in in vivo histological analysis.

In the chemical interface field, Xiong et al. conjugated Lactoferrin with BP on a Parkinson’s model drug to enable targeted brain delivery, as shown in Figure 4E [23]. The chemical interface significantly increased blood–brain barrier (BBB) permeability both in vitro and in vivo, thereby providing notable anti-Parkinsonian effects. In addition, the biocompatibility of this conjugate has been demonstrated through cellular and in vivo experiments. Liu et al. constructed a multifunctional hydrogel system mixed with black phosphorus–methylene blue based on the Schiff base reaction, targeting Alzheimer’s disease by inhibiting tau phosphorylation, modulating mitochondrial function, and alleviating neuroinflammation [92], as shown in Figure 4F. Tan et al. successfully developed a novel drug-delivery interface of black phosphorus nanosheets (BP) modified with the base-targeting peptide RVG29 and combined with Hypericin (HYP), demonstrating that the interface significantly alleviates depressive-like behaviors and oxidative stress in mice, which could create new avenues for the treatment of depression, as shown in Figure 4G [93]. Qian et al. developed a BP/polycaprolactone (PCL) nanoscaffold using concentrically integrative layer-by-layer bioassemblies, as shown in Figure 4H [94]. The nanoscaffold induced angiogenesis and neurogenesis and stimulated calcium-dependent axon regrowth and remyelination under conditions of mild oxidative stress. The outstanding neural regenerative capacity was attributed to the superior electrical conductivity of the BP nanoplates within the scaffolds. Furthermore, the scaffolds provided long-term mechanical stability and exhibited biocompatibility in vivo experiments.

### 3.3. Hexagonal Boron Nitride (h-BN)

h-BN is structurally more similar to traditional graphene and is often referred to as “white graphene”. The building blocks of monolayer h-BN exhibit a hexagonal structure consisting of equal numbers of boron and nitrogen atoms. Unlike other famous layered structures such as graphite, which are either fully conductive or semi-conductive, partially covalent bonds between B and N atoms disturb the electronic-state symmetry through the narrowing of the sp2-derived π bands and generate a large band gap (∼5.1–5.9 eV) in h-BN [64]. Although its electrical insulation is disadvantageous for electron transport, the high surface area, active edges, low toxicity, and catalytic properties of h-BN make it suitable for improving chemical interfaces.

In the realm of chemical interfaces, Ouyang et al. developed an innovative dopamine (DA) detection interface based on carbon-doped hexagonal boron nitride (C-hBN) [32]. In terms of electrochemical performance, the C-hBN modified glassy carbon electrode (C-hBN/GCE) exhibited a larger electroactive surface area (EASA) of 0.26 cm² and a lower electron transfer resistance (Rct) of 509 Ω, representing a significant improvement over the pure glass electrode, which had an EASA of 0.21 cm² and an Rct of 1875 Ω. The defects introduced on the C-hBN/GCE provided more active sites for DA and facilitated electron transfer. Additionally, carbon doping was found to reduce the band gap of h-BN, thereby enhancing electron transfer between the electrode and the solution. The developed sensor demonstrated excellent analytical performance in DA detection, with a low detection limit of 5.8 nM, a wide linear concentration range of 0.01–40 µM and 40–300 µM, high sensitivity of 2037 µA⋅mM⁻¹⋅cm⁻², and outstanding selectivity. In addition, Zeybekler developed an HBN-PDA/Anti-T-Tau/SCPE immunosensor capable of detecting T-Tau in blood serum [95]. Based on the nanocomposite design, the sensor exhibited outstanding electrochemical activity for T-Tau detection with a wide linear range, good selectivity, and a trace-level detection limit.

In addition, due to the electrical insulation properties of h-BN, it can also be used as an electrically isolated layer. Lee et al. developed an ultra-thin surface electrode array neural probe, NeuroWeb, based on the ultra-thin thickness and rational design of the open lattice structure with h-BN isolated layers, as shown in Figure 5 [96]. The probe combines the advantages of an implantable multi-electrode array for unit spike detection with the minimally invasive nature of a surface electrode array (SEA). The electrode exhibits superior electrochemical properties compared to the standard polymer-metal SEA (SU8-Au SEA). Additionally, the 30-channel electrodes showed minimal variation in average amplitude between days 1 and 7, enabling long-term recording and monitoring of neural activity at the single-unit level.

### 3.4. 2D Metal-Organic Frameworks (MOFs) and Covalent Organic Frameworks (COFs)

Metal-organic frameworks (MOFs), which are constructed through the self-assembly of metal ions/clusters and organic linkers, are a category of crystalline hybrid materials [97]. Thousands of MOFs are discovered every year due to the wide variety of metal ions and polymer linkers. MOFs have electrochemical properties of interlayer anisotropy, and this electroanalytical advantage can be applied to the electrochemical detection of neurochemical substances. MOFs own obvious superiorities as follows [98]: (a) relatively weak coordination bonds endow MOFs with a merit of degradability in complex physiological environments, which permits the function of a stimuli-responsive drug-release system while avoiding the long-term accumulation and potential toxicity in vivo. (b) Large pores/channels and a high porosity of MOFs facilitate the high payload capacity of drug-delivery systems. (c) The tunability of MOFs facilitates the realization of multifunctionality. In addition, compared with one- and three-dimensional aggregates, two-dimensional MOFSs have superior properties due to shorter transport distances during ion diffusion and charge transfers. Stolz et al. reported that the unique crystallographic orientation of 2D MOFs can be used to control the fundamental interactions between the analyte and the electrode [33]. The {001} planes were advantageous in adsorption-driven electrochemical processes, such as the oxidation of DA. The integration of {001} oriented Ni3(HHTP)2 onto the GCE allowed for the detection of DA concentrations as low as 9.9 ± 2 nM in PBS, as shown in Figure 6A. The modified electrode exhibited excellent sensitivity and selectivity towards DA in simulated cerebrospinal fluid. Wang et al. used conjugated molecular wires to modulate MOF microelectrodes for monitoring dopamine in the mouse brain [99]. Conductive Ni₃(HHTP)₂ was modified on the electrode surface with 4-(thiophen-3-ylethynyl) benzaldehyde (RP1). The rational molecular length and the conjugated structure endowed RP1 with an efficient tunneling capability to specifically decrease Gibbs free energy of DA oxidation, leading to a high selectivity for DA determination. Additionally, implantation results in the mouse brain demonstrated that the microelectrodes exhibited outstanding biocompatibility, with negligible brain tissue inflammation caused by the implantation.

The structural diversity of two-dimensional metal–organic frameworks (MOFs) may inspire researchers to explore their potential in biomedical electronics. However, the practical implementation of 2D MOFs is limited by their poor conductivity. Beyond MOF structures, researchers have also identified that organic molecules can produce lightweight, porous crystalline covalent organic frameworks (COFs) with superior charge-transport capabilities. COFs are typically composed of light atoms such as C, N, O, and H, which generally exhibit lower toxicity and higher biocompatibility compared to MOFs. Zhou et al. demonstrated that 2,4,6-trihydroxybenzene-1,3,5-tricarbaldehyde (Tp)/ p-phenylenediamine (Pa) COFs can provide the carbon fiber microelectrode an excellent in vivo performance with enhanced antibiofouling and antichemical fouling ability, selectivity, and stability. The electrodes modified with thin-layer TpPa COF were used to measure and analyze dopamine in the brains of model mice with Parkinson’s disease (PD) [100]. The physicochemical properties of TpPa COF endow the electrodes with excellent selectivity for dopamine (DA), limiting DA electro-oxidation and preventing the aggregation of DA and its oxidation products on the electrode. Additionally, the modified electrodes exhibit anti-biofouling capabilities, excellent anti-chemical fouling properties, and high in vivo stability, as shown in Figure 6B When implanted into the striatum of live mice, the electrodes showed no signs of glial cell aggregation or brain tissue damage over 21 days.

### 3.5. MXene

Since the first discovery of MXenes in 2011, their exceptional electrical conductivity and biocompatibility have received significant attention [101]. MXenes are novel two-dimensional layered materials consisting of nanostructures formed from early-transition metal carbides, nitrides, and carbonitrides [102]. These materials are synthesized through the selective etching of A-element layers from an MAX phase, resulting in the creation of two-dimensional nanosheets with distinctive electronic properties and high-surface areas. The MAX phase is characterized by the general formula Mₙ₊₁AXₙ (*n*= 1, 2, 3), where M stands for transition metals from the d-block (such as Ti, Sc, C, etc.), A stands for main-group sp elements, and X stands for carbon or nitrogen elements [103]. Most MXenes exhibit metal-like electron-transfer properties and can be made semiconducting by changing the transition metal type and the structure of the M-site, as shown in Figure 7A [104]. Additionally, the surface of MXenes can be functionalized by modifying the synthesis process to introduce various terminal functional groups. For instance, the presence of hydroxyl or oxygen groups on the surface renders MXenes hydrophilic [105]. Due to their excellent properties, MXenes such as Ti₃C₂ have been widely studied for applications in neural interfaces.

Flavia Vitale et al. were the first to apply MXene materials in implantable neural microelectrodes. Using the high-throughput processes with standard photolithographic patterning, they fabricated micro-patterned Ti3C2 MXene film, as shown in Figure 7B [106]. These electrodes exhibited an interface impedance approximately four times lower than comparable gold contacts and improved the SNR for in vivo neuronal spike recording. Subsequently, they utilized laser patterning and Ti3C2 MXene ink to fabricate multichannel neural arrays [107]. The electrochemical properties of MXene electrodes (CSCc = 1495.3 ± 60.5, CICc = 2.01 ± 0.46) are significantly superior to those of platinum electrodes (CSCc = 4.4 ± 0.7, CICc = 0.05 ± 0.001). The recording capabilities of these electrodes have been validated in dry electroencephalography (EEG), high-density surface electromyography (HDsEMG), and electrocardiogram (ECG) recordings in humans. Additionally, their efficacy as brain electrodes was further tested through electrocorticography (ECoG) recordings in pigs and in vivo cortical stimulation in rats. Notably, the electrodes also demonstrated compatibility with MRI and CT imaging.

In addition to providing outstanding electrical properties, MXene can also contribute to improving the optical and magnetic behavior of interfaces. Song et al. reported a ≈20 nm-thick MXene film (11,000 S cm^−1^@89%) crosslinked by poly (3,4-ethylene dioxythiophene): polystyrene sulfonate (PEDOT: PSS) as an electronic tattoo (MPET), as shown in Figure 7C [21]. The extraction of intercalated water molecules, combined with the cross-linking of PEDOT, forms a tightly packed MXene layered structure, resulting in an exceptional combination of electrical conductivity and transparency. Furthermore, the film can detect electrophysiological signals and perform imaging by MRI and fNIRS with minimal interferences of the magnetic and optical field, enabling multimodal cognitive neural monitoring during long-term application.

It is noteworthy that MXene can also be utilized in chemical interfaces, as MXene can detect dopamine molecules through changes in conductivity via π–π interactions. Xu et al. developed a highly sensitive device based on Ti3C2–MXene–FET for detecting neurotransmitters and probing action potentials in primary hippocampal neurons, as shown in Figure 7D [108]. These transistors enable highly sensitive label-free detection of dopamine with a detection limit as low as 100 × 10^−9^ M. A proper bandgap in MXenes is thought to help improve sensitivity by reducing current leakages in the off state.

To enhance the sensitivity and stability of the chemical interface, composite responsive structures consisting of MXene nanosheets and other nanomaterials have been researched. Xue et al. prepared Ti3C2Tx/PtNPs using a solution system and freeze-drying method and developed a DA sensor based on the composite structure [109]. The resulting Ti3C2Tx/PtNPs/GCE shows a good electrochemical response for the detection of DA and UA due to its large surface area, low electron-transfer resistance, and negatively charged surfaces. Amara et al. provided an economical and highly stable MXene-based sensing interface [110]. After uniformly depositing 1-methyl imidazolium acetate ionic liquid (IL)-MXene to modify graphitic pencil electrode (GPE), the sensitivity of the IL-MXene/GPE for DA was found to be 9.61 µA µM^−1^ cm^−2^. The developed sensor also showed good reproducibility and practicability in human serum.

Additionally, due to the colloidal behavior of 2D MXene, it has also been explored for constructing novel non-invasive neural interfaces. Sha et al. have presented a method for assembling neural interfaces directly within tissues without the need for polymer carriers or surgery, as shown in Figure 7E [111]. This assembly is delivered to specified nerve terminals via a jet injector. They utilize active oxygen catalysis-doped poly(3,4-ethylenedioxythiophene) to enhance π–π interactions between nanosheets, producing a conductive and biodegradable interface. This interface effectively modulates local immune activity, promoting sensory and motor nerve-function recovery in mice with nerve injuries. Moreover, it engages the vagus-adrenal axis in freely moving mice, inducing a catecholamine neurotransmitter release and suppressing systemic cytokine storms. This innovative strategy selectively targets neural substructures, supporting local and systemic immune modulation.

**Figure 7 ijms-25-08615-f007:**
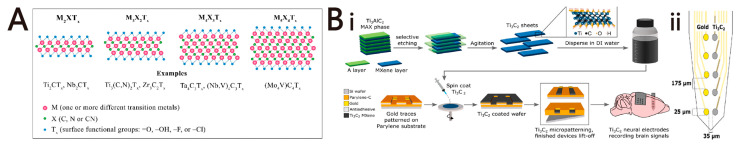
(**A**) Schematic representation of the structure of MXenes with *n* = 1–4 and some examples. Reprinted with permission from [104] Copyright © 2021, The American Association for the Advancement of Science. (**B**) Synthesis, characterization, and application of Ti3C2 for neural recording. (i) Schematics illustrating synthesis and atomic structure of Ti3C2, fabrication of Ti3C2 neural microelectrode arrays, and application of Ti3C2 arrays for recording brain activity at different locations in the rat brain. (ii) Ti3C2/Au intracortical electrode array. Reprinted with permission from [106] Copyright © 2018, American Chemical Society. (**C**) Design and preparation of a highly stable, transparent, and conductive MP film. (i) Schematic illustration of the chemical structure of MP film (MXene film crosslinked by PEDOT: PSS). (ii) Multimodal cognitive monitoring by MPET. Reprinted with permission from [21] Copyright © 2023, Wile. (**D**) Schematic of a biosensing device based on MXene field-effect transistors. Reprinted with permission from [108] Copyright © 2016, Wiley. (**E**) Jet-injected neural interface promoted the recovery of motor and sensory nerves in peripheral nerve injury and drove the vagal–adrenal anti-inflammatory axis with low-intensity ES in awake mice. Reprinted with permission from [111] Copyright © 2023, PNAS.

### 3.6. Transition Metal Dichalcogenides (TMDs)

The chemical formula of transition metal sulfur compounds (TMDs) can be expressed as MX2 (M stands for IV B~X B group transition metal elements, X stands for S, Se, and Te). The different structures of 2D TMDs depend on the stacking pattern of the three layers of atoms, with the two most common structures being the trigonal 2H phase and the octahedral 1T phase. The 2H phase can be regarded as an A-B-A stacking in which chalcogen atoms in different layers occupy the same position, while the 1T phase can be regarded as a centrosymmetric structure composed of ABC stacking mode [112]. In addition to the 2H and 1T phases, another common structure is the 1T’ phase. This structure can be viewed as a periodically twisted 1T phase, where two neighboring metals are close to each other and form a chain of metal sawtooths, as shown in Figure 8A [113,114]. At the same time, many physical and chemical properties of TMDs are closely related to their phases; for example, 2H-MoS2 is semiconducting, whereas 1T-MoS2 exhibits metallic properties, providing various opportunities to realize different properties [115]. TMDs demonstrate many interesting properties for the neural interface. These materials are available in both metallic and semiconducting forms, with bandgaps ranging from 1.5 to 3 eV. Their energy band structure can be effectively tuned by adjusting the number of layers, applying strain, or employing doping techniques [116]. In terms of physical properties, TMDs exhibit excellent mechanical flexibility and high fracture resistance. The fracture strength of monolayer TMDs not only exceeds that of black phosphorus (BP), but its Young’s modulus of about 200 GPa is also higher than that of most other 2D materials [117]. Its unique layered structure is also able to be realized by non-covalent interactions such as gold-mercapto bonding, π–π stacking, and liganding using sulfur vacancies to efficiently absorb a variety of molecular modifications. These interactions and linkages offer a wide range of possibilities for the functionalization of TMDs [118].

In the field of optical interface, Choi et al. developed a high-density, ultra-soft MoS2-graphene photodetector array that can activate the optic nerve based on incident light and be used as a retinal prosthesis, as shown in Figure 8B [37]. The photoresponsivity of MoS2-graphene phototransistors is two-to-three orders of magnitude higher than that of silicon phototransistors of the same size due to the efficient light absorption of MOS2. In addition, since MOS2 has a wide bandgap and does not absorb the infrared spectrum, the IR filter can be removed from the device design to reduce the device thickness and increase flexibility.

The adsorptive properties of TMDs can synergistically enhance the detection capabilities of chemical interfaces. Yang et al. proposed the fabrication of a degradable interface for detecting deep-brain neurotransmitters and neurophysiology by coating a silicon-based interface with a solution-processable heterostructure of 2D TMDs (MoS₂ and WS₂) and iron (Fe) nanoparticles, as shown in Figure 8C [119]. The primarily 1T-phase TMDs not only exhibit high electrical conductivity, enabling effective charge transfer, but also provide numerous active sites for the attachment of Fe-NP catalysts, synergistically enhancing dopamine detection. The two types of coatings, based on MoS₂ or WS₂, offer options for short-term or long-term monitoring tools and both demonstrated good biocompatibility in rat brain histological analysis.

TMDs are also widely used in electrical interfaces. Gunapu et al. reported the electrochemical deposition of poly(3,4-ethylenedioxythiophene) (PEDOT) and molybdenum disulfide (MoS₂) composite films on a custom 32-channel gold microelectrode array (MEA), as shown in Figure 8D [120]. The composite films significantly reduced electrode impedance, enhanced charge-storage capacity, and increased the average charge injection capacity while exhibiting high electrochemical stability. Additionally, the coating demonstrated good biocompatibility in L929 cell experiments. Kireev et al. fabricated nanoscale thickness PtSe₂ and PtTe₂ electronic tattoos using a thermal-assisted conversion (TAC) CVD process, as shown in Figure 8E [121]. These platinum-based TMD electronic tattoos significantly outperform traditional gold and graphene electronic tattoos in terms of thin-film resistance, skin-contact performance, and electrochemical impedance. Their performance is comparable to that of medical-grade silver/silver chloride gel electrodes. Notably, platinum tattoos exhibit superior electrical performance compared to monolayer graphene tattoos. Among them, PtTe₂ tattoos demonstrate even better performance than PtSe₂ tattoos, attributed to the higher metallicity of PtTe₂.

In addition, the rational use of van der Waals (VDW) interactions in 2D TMDs can construct interfaces with excellent mechanical properties. Yan et al. fabricated highly stretchable, adaptive, conformal, and breathable thin-film electronic devices by spin-coating molybdenum disulfide (MoS₂) nanosheet ink, as shown in Figure 8F [122]. These devices, prepared through van der Waals interactions without bonding, can seamlessly integrate with biological tissues to form electronic-biological hybrids. Nanosheets in van der Waals (VDW) interfaces, where no chemical bonding is present, can slide and rotate to accommodate local tension or compression, thus preserving the VDW interface structure and conductive pathways. VDW thin-film skin-gate transistors (VDWTFs) exhibit a cutoff frequency of approximately 100 kHz, which is sufficient for monitoring most electrophysiological signals from the human body. Furthermore, precise signals were recorded during EEG and ECG monitoring processes.

## 4. Conclusions and Future Perspectives

In this review, we discuss the challenges faced by neural interfaces, including sensitivity, thermal issues, and biocompatibility, and summarize parameters that can be used to evaluate these challenges. Specifically, chemical and optical interfaces require highly specific evaluation compared to electrical interfaces. Therefore, when dealing with chemical and optical interfaces, it is essential to design specialized evaluation systems based on their requirements and optimize interface structures accordingly. For electrical interfaces, it is significant to consider capacitance behavior, impedance, and results from biocompatibility testing.

Recent research has demonstrated that the judicious use of 2D nanomaterials can effectively enhance and expand the functionalities of neural interfaces, exhibiting unique advantages and challenges. Graphene and GBMs have shown excellent performance in electrical, mechanical, and optical properties, significantly improving electrical and optical neural interfaces. However, challenges remain in their biocompatibility and fabrication processes that require further refinement. Black phosphorus, known for its optical properties and superior biological behavior, has been applied in optothermal stimulations and neural drug delivery. Its limited application in neural interfaces may stem from its susceptibility to oxidation and inferior electrical conductivity compared to other 2D materials. Notably, significant research progress has been made in high-performance black phosphorus electrodes in traditional capacitors and electrode fields. With advancements in miniaturization and electrode biocompatibility, the realization of black phosphorus neural electrodes may be not far. The ability to functionalize and the biocompatibility of h-BN lead to its widespread use in chemical interfaces for specific recognition of neurotransmitters. Notably, pristine h-BN is insulating and is often modified to enhance its electrical properties before being used in electrochemical interfaces to detect neurotransmitters. While its bandgap restricts its role as an electrode in electrical interfaces, highly insulating and stable h-BN can serve as an excellent insulating layer. Although MOFs and COFs are slightly inferior in terms of electrical properties, their highly specific surface area, tunability, and unique pore structure enable them to provide unique advantages in chemical interfaces such as the catalysis, electrochemical sensing, and drug delivery. The excellent electrochemical properties of MXenes provide the interface very low impedance and excellent CSCC. The surface can be functionalized to further enrich the interface function. However, it is prone to oxidative failure and lacks long-term reliability, requiring protection through coatings or the construction of more stable composite structures with other materials. The excellent mechanical properties, tunable band gap, and good electrochemical properties of TMDs enable them to perform well in both electrical and optical interfaces. And their biological behavior can be further improved by surface functionalization. However, similar to MXenes, they are easily oxidized.

Furthermore, the complex biocompatibility of 2D nanomaterials should be a key focus. The biocompatibility and nanotoxicity of 2D nanomaterials may not be directly inferred from their corresponding bulk counterparts, as the nano–bio interactions between 2D nanomaterials and biological entities are unique and heavily dependent on numerous physicochemical parameters of the nanomaterials, including lateral size and its distribution, surface area, shape, concentration, stability, number of layers, surface charge, chemical composition, purity, and the surface functionalization of nanomaterials [123]. This complexity means that similar 2D nanomaterials may exhibit significantly different biocompatibility in similar studies. Additionally, even when using completely safe 2D nanomaterials as raw materials, variations introduced during processing can lead to suboptimal biocompatibility in the final device. Therefore, testing for biocompatibility is still essential.

To meet increasing functional demands of neural interfaces, enhancing performance through strategies such as functional group modification, doping, and composite structure construction has become the mainstream focus of research. Considering the diverse demands on electrical, optical, and electrochemical performance across various neural interfaces, targeted optimizations are essential. Future research is likely to focus on improving the fabrication or modification methods for 2D nanomaterial electrodes, utilizing fundamental insights such as nanoarchitectonics [124].

## Figures and Tables

**Figure 1 ijms-25-08615-f001:**
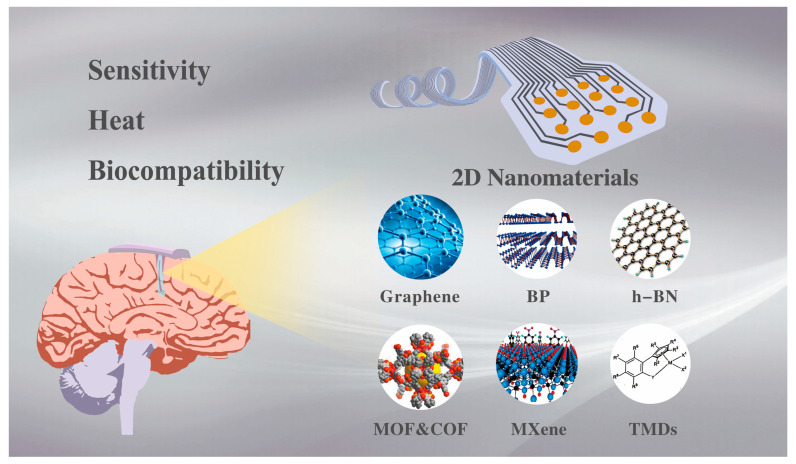
Challenges for neural interfaces and common 2D nanomaterials for the applications.

**Figure 2 ijms-25-08615-f002:**
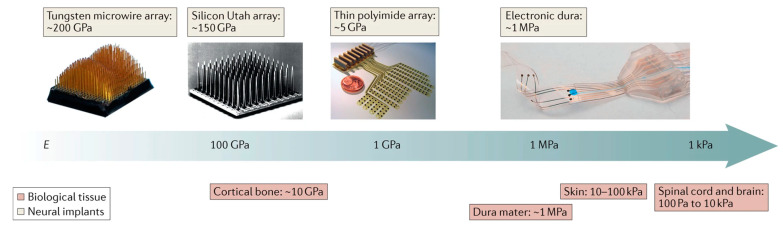
Mechanical mismatch between neural tissue and artificial implanted electrodes.

**Figure 3 ijms-25-08615-f003:**
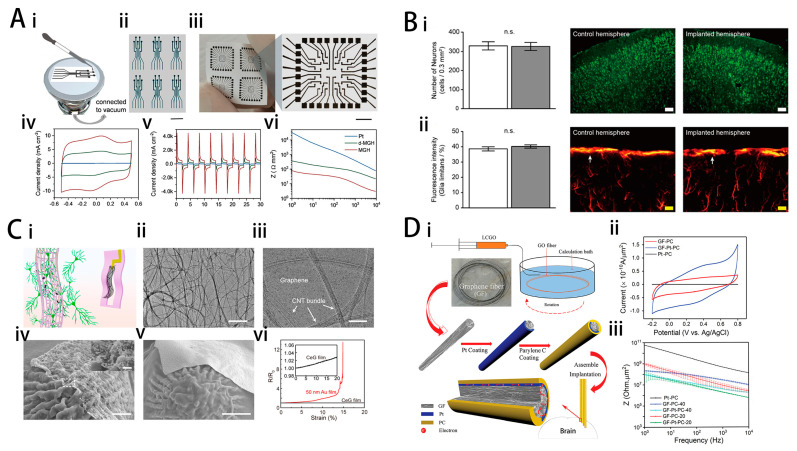
(**A**) (i) Schematic of mask-assisted filtration for microelectrode fabrication. (ii, iii) Photographs of MGHbased cuff microelectrodes and planar microelectrode arrays. The scale bars in (A–ii) and (iii) are 800 and 500 µm, respectively. (iv) Cyclic voltammetry measurements of MGH (red), d-MGH (green), and Pt (blue) microelectrodes measured at a scan rate of 10 mV s^−1^. (v) Current densities of MGH, d-MGH, and Pt microelectrode measured at 500 Hz frequency, ±0.5 V bipolar pulses. (vi) Bode plots for the impedance of MGH, d-MGH, and Pt microelectrodes. The electrolyte is PBS. Reprinted with permission from [71] Copyright © 2022, Wiley. (**B**) (i) No difference was observed in the number of neurons (labeled with NeuN, a neuronal molecular marker) between implanted (gray) and control (white) hemispheres of awake animals (scale bar = 100 μm). (ii) No difference was observed in the number of glia (labeled with GFAP, a glia molecular marker) between implanted (gray) and control (white) hemispheres of the awake animals (scale bar = 20 μm). The arrows indicate glia limitans, the index for brain inflammation. Data are expressed as the mean ± SEM. n.s. not significant. Reprinted with permission from [80] Copyright © 2022 Springer Nature. (**C**) (i) Schematic illustrating the design of CeG_MEAs for neural interfacing. (ii, iii) TEM images of the CeG film. Scale bars: 500 nm in (ii) and 50 nm in (iii). (iv) SEM images of a PtNPs-coated CeG nanofilm conformally spreading on a rose petal replica with tightly packed micropapillae and nanowrinkles. Scale bar: 2 μm. Inset scale bar: 500 nm. (v) SEM image of a 50 nm-thick gold film floating on a rose petal replica. Scale bar: 5 μm. (vi) Relative resistance changes of a CeG film and a 50 nm-thick gold film under the tensile strain from 0 to 20%. Reprinted with permission from [20] Copyright © 2023 American Chemical Society. (**D**) (i) Process of GO fiber preparation, GF-Pt microelectrode fabrication, and intracortical implantation. (ii) CVs of the microelectrodes at 10 mV s^−1^ in PBS solution. (iii) Modulus of impedance of microelectrodes. Reprinted with permission from [81] Copyright © 2019, Wiley. (**E**) (i) A schematic drawing of the DBS–fMRI study using GF bipolar microelectrodes. (ii) A representative SEM image of the axial external surface of a GF fiber. Inset, magnified image of the region in the dashed box. Scale bar, 20 μm; inset, 5 μm. (iii) The picture of a GF bipolar microelectrode assembly. Inset, SEM image of the GF bipolar microelectrode tip, showing two GFs (bright core) with each one insulated with Parylene-C film (dark shell). Scale bar, 1 cm; inset, 100 μm. (iv) Representative coronal (left) and horizontal (right) sections of the T2 MRI images of rat brains implanted with a GF bipolar microelectrode. Reprinted with permission from [22] Copyright © 2022 Springer Nature. (**F**) Normalized power spectrum of the recorded signal from Au and graphene electrodes (zoomed-in) after shining light with 10 Hz frequency. Reprinted with permission from [38] Copyright © 2022 Springer Nature.

**Figure 4 ijms-25-08615-f004:**
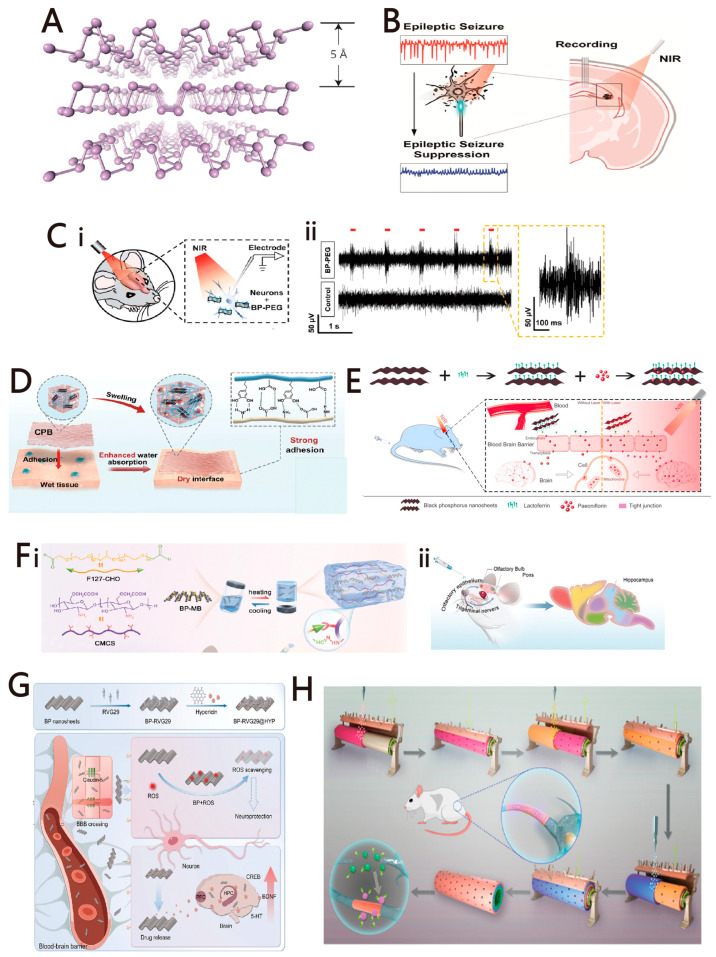
(**A**) Atomic structure of black phosphorus. Reprinted with permission from [84] Copyright © 2014, Springer Nature Limited. (**B**) Schematic representation of BP flake-enabled neuromodulation for suppressing electrical signals in epilepsy. Reprinted with permission from [89] Copyright © 2024, American Chemical Society. (**C**) In vivo neural stimulation using BP–PEG nanosheets. (i) Schematic of BP-based neural stimulation and extracellular recording in brains of anesthetized animals. (ii) Representative recordings of spiking activities in cells in response to pulses of NIR light with/without BP–PEG nanosheets. Reprinted with permission from [90] Copyright © 2021 Wiley. (**D**) Wet adhesion of CPB. With a strong swelling capacity when adhered to wet tissues, the interfacial water can be reduced or even removed. The interaction between the patch and the tissue is not affected by water molecules and finally results in stronger adhesion of CPB. Reprinted with permission from [91] Copyright © 2024, The Author(s). (**E**) A schematic overview of the development and use of the Lf-BP-Pae nano-platform for the treatment of PD. Reprinted with permission from [23] Copyright © 2020 Elsevier Ltd. All rights reserved. (**F**) BP-MB@Gel application for improving AD pathology. (i) The preparation of BP-MB@Gel. (ii) Schematic illustration of the IN administration of BP-MB@Gel. Reprinted with permission from [92] Copyright © 2024 Wiley. (**G**) A schematic overview of the development and use of the BP-RVG29@HYP nano-platform for the treatment of depression. Reprinted with permission from [93] Copyright © 2024 Wiley. (**H**) Schemes of concentrically integrative layer-by-layer bioassembly (CI-LBLB) of a BP/polycaprolactone (PCL) nanoscaffold. Reprinted with permission from [94] Copyright © 2019, American Chemical Society.

**Figure 5 ijms-25-08615-f005:**
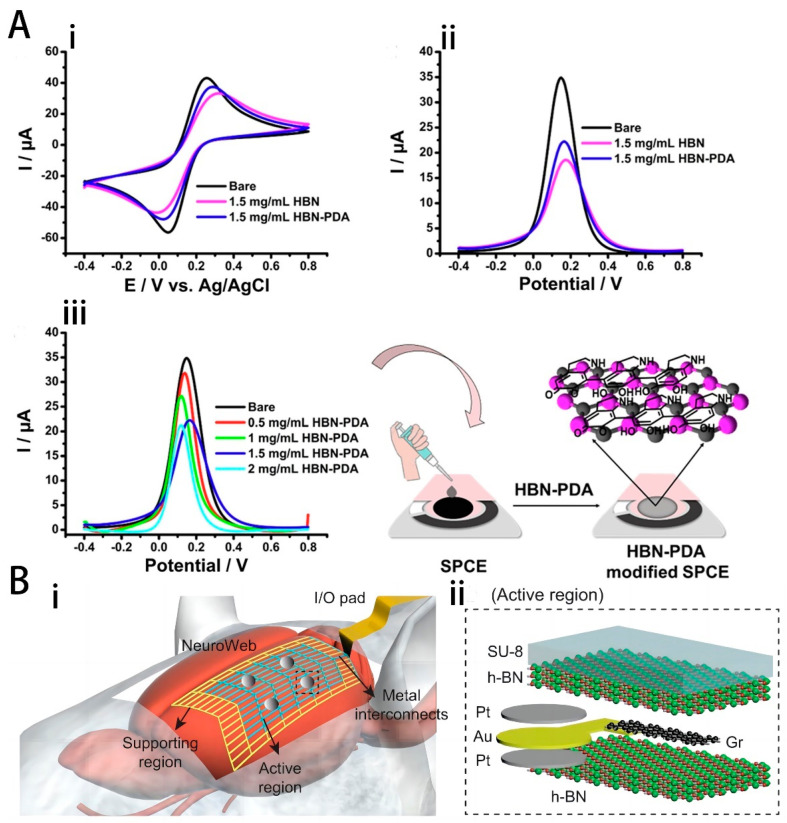
(**A**) (i) CV profiles of bare SPCE, SPCE/HBN, and SPCE/HBN-PDA at 1.5 mg/mL nanocomposite concentration, (ii) DPV profiles of bare SPCE, SPCE/HBN, and SPCE/HBN-PDA modified SPCEs at 1.5 mg/mL nanocomposite concentration, (iii) DPV profiles of SPCE/HBN-PDA using various concentrations of HBN-PDA, (all measurements were carried out in in 10 mM pH 7.4 PBS contained 5.0 mM HCF and 0.1 M KCl, scan rate: 50 mV/s). Reprinted with permission from [95] Copyright © 2023, Elsevier. (**B**) (i) Schematic illustration of NeuroWeb on the mouse brain surface. NeuroWeb consists of the active region, supporting region, and metal interconnects. The detected action potentials using NeuroWeb are sent to an external interface via I/O pads. (ii) Close-up view of the black dashed box in (ii), which consists of the Pt recording electrode (gray), Au joint (yellow), Gr line (black), top and bottom h-BN insulating layers (green), and SU-8 (light blue). Reprinted with permission from [96] Copyright © 2023, PubMed Central.

**Figure 6 ijms-25-08615-f006:**
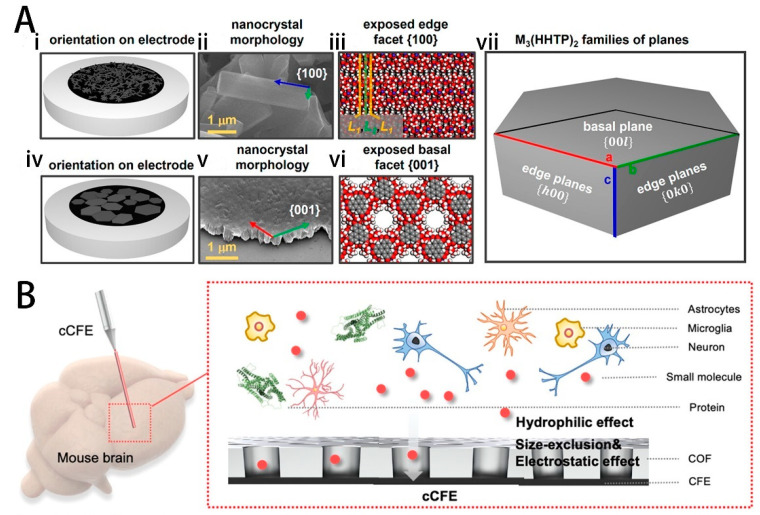
(**A**) M3(HHTP)2 in the form of (i) hydrothermally grown and drop-casted MOF and (iv) MOFs obtained by epitaxial growth and Langmuir–Blodgett deposition with their anticipated orientation on the electrode surface. Scanning electron micrographs of (ii) nanorods and (v) nanosheets. Idealized models of exposed crystal facets belonging to the Bragg family of planes (iii) {100} or (vi) {001} were generated from reported crystal structures (blue: metal (Ni or Co), gray: carbon, red: oxygen, and white: hydrogen). (vii) A representative hexagonal nanocrystal with overlaid families of planes. Reprinted with permission from [33] Copyright © 2023, American Chemical Society. (**B**) Schematic illustration of the antibiofouling mechanism of cCFE. Reprinted with permission from [100] Copyright © 2022, American Chemical Society.

**Figure 8 ijms-25-08615-f008:**
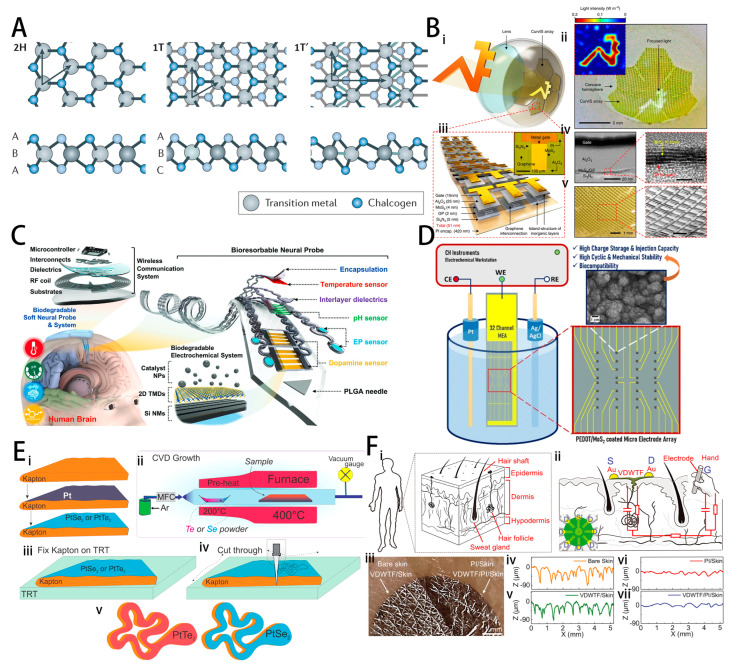
(**A**) Atomic structure of single layers of transition metal dichalcogenides (TMDCs) in their trigonal prismatic (2H), distorted octahedral (1T), and dimerized (1T′) phases. Reprinted with permission from [113] Copyright © 2017, Springer Nature. (**B**) (i) Schematic illustration of the high-density CurvIS array based on the MoS2-graphene heterostructure. (ii) Optical camera image of the high-density CurvIS array. Inset shows the image (i.e., university logo) captured by the CurvIS array. (iii) Schematic illustration of the device design. Inset shows an optical microscope image of a single phototransistor. (iv) Cross-sectional transmission electron microscope image of the MoS2-graphene phototransistor (left) and the magnified image of the MoS2-graphene heterostructure (right). (v) Optical (left) and magnified scanning electron microscope (right) image of the high-density CurvIS array on the concave hemisphere. Reprinted with permission from [37] Copyright © 2017, Springer Nature. (**C**) Schematic illustration of a biodegradable, implantable electrochemical brain-integrated system for investigating neurotransmitters with associated variations in pH, temperature, and EP signals. Reprinted with permission from [119] Copyright © 2022, John Wiley and Sons. (**D**) Schematic of PEDOT/MoS2 coated micro electrode array. Reprinted with permission from [120] Copyright © 2020, Springer Nature. (**E**) Schematic overview of Kapton-based Pt-TMD tattoo design and fabrication flow. (i) Evaporation of thin Pt on top of the Kapton film, followed by TAC conversion into Pt-TMD. (ii) Schematic of the TAC CVD process. (iii) Post-CVD growth, the Pt-TMD/Kapton sample is fixed on top of a TRT. (iv) Mechanical patterning process of the Pt-TMDs grown on Kapton film. (v) Schematics of the final PtSe2 and PtTe2 tattoos supported by Kapton. Reprinted with permission from [121] Copyright © 2021, American Chemical Society. (**F**) (i) The structure of human skin. (ii) Schematic of a skin-gate VDWTF transistor with Au source and drain electrodes and an iron rod gate electrode held by a human subject. (iii) Photograph of the freestanding VDWTF on a replica of human skin (left) and the VDWTF supported by a 1.6 μm-thick polyimide substrate on a replica of human skin (right). (iv–vii) Height profiles corresponding to the line scan in different areas of (iii). Reprinted with permission from [122] Copyright © 2022, The American Association for the Advancement of Science.

**Table 1 ijms-25-08615-t001:** Methods for assessing the biocompatibility of neural interface.

Test Name	Description
Cytotoxicity Test	Assesses whether the material negatively affects cell viability and proliferation capacity through in vitro experiments such as MTT assays and LIVE/DEAD staining.
Cell Adhesion and Proliferation Test	Evaluates cell adhesion and growth capacity on the material surface through in vitro experiments.
Inflammatory Response Assessment	Evaluates the degree of inflammatory response triggered by the material using an animal model. Common parameters include cytokine levels and immune cell infiltration surrounding the implant.
Corrosion and Degradation Test	Weight changes, structural damage, or changes in the chemical composition of the implant after corrosion in simulated body fluid are used to assess durability.
Chronic Implantation Study	Observes the interaction between the implant and the surrounding tissue in an animal model, including scar formation, tissue encapsulation, and chronic inflammatory response.

## Data Availability

Not applicable.

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
