# Peer review of "Applications of 2D Nanomaterials in Neural Interface"

_ijms, 2024, doi:10.3390/ijms25168615_

Round 1

Reviewer 1 Report

Comments and Suggestions for Authors

The work explores how advanced materials like graphene and MXenes are revolutionizing the way we connect neural tissues to devices. These materials help overcome major hurdles such as improving the sensitivity of these connections, managing heat, and ensuring they are safe and compatible with the human body. The authors highlight how these materials can make neural interfaces more precise and functional, enabling better MRI scans and targeted drug delivery. The review highlights the need for ongoing innovation and testing of these materials to keep advancing this field.

Below my comments: 

1. While the review touches on the biocompatibility of 2D nanomaterials, it could benefit from a deeper dive into this topic. Including more detailed studies on how these materials interact with neural tissues over the long term would be valuable. Focusing on immune responses and potential chronic effects would provide a fuller picture of their safety and effectiveness.

2. The review does a great job highlighting the benefits of various materials, but it would be even better with more detailed comparisons backed by data. Adding specific numbers on things like signal-to-noise ratio, impedance, and charge injection capacity would make it easier to see how these materials stack up against each other. 

3. The section on heat management is quite brief and could use some more detail. It would be really helpful to explore innovative ways to manage heat in neural interfaces, such as new cooling techniques or material modifications. 

4. The review covers a wide range of neural interfaces, but it doesn't go into much detail about the specific challenges of each application. Including case studies or specific examples where 2D nanomaterials have been successfully used in particular applications, like deep brain stimulation or cortical recording.

5. The section on future research directions is a bit general and could be more specific. It would be helpful to outline concrete research questions and detailed experimental setups to address current gaps in knowledge. 

Comments on the Quality of English Language

The English in the manuscript is generally quite good, but some areas could use a little polishing. Some sentences are long and complex, which can make them a bit tough to follow. Breaking these into shorter, clearer sentences would help a lot. There are also a few grammatical errors and awkward phrases that could be smoothed out with a careful check. 

Author Response

We sincerely appreciate the valuable feedback provided by the reviewer. We found these suggestions to be very helpful, and we have carefully addressed each point. Below are our responses:

1. While the review touches on the biocompatibility of 2D nanomaterials, it could benefit from a deeper dive into this topic. Including more detailed studies on how these materials interact with neural tissues over the long term would be valuable. Focusing on immune responses and potential chronic effects would provide a fuller picture of their safety and effectiveness.

In section 2.3, we have added a discussion on the typical immune response that occurs when interface materials are introduced into tissues. This addition aims to enhance the understanding of the safety improvements discussed later in the manuscript.

2. The review does a great job highlighting the benefits of various materials, but it would be even better with more detailed comparisons backed by data. Adding specific numbers on things like signal-to-noise ratio, impedance, and charge injection capacity would make it easier to see how these materials stack up against each other. 

We have included electrical performance parameters for the MXene conductive interfaces in section 3.5. However, due to the various and unique properties of chemical and optical interface materials, it is difficult to represent these characteristics numerically, so we have not included quantitative numbers for them.

3. The section on heat management is quite brief and could use some more detail. It would be really helpful to explore innovative ways to manage heat in neural interfaces, such as new cooling techniques or material modifications. 

We have significantly expanded our discussion on the impact of thermal management in section 2.2. We agree that thermal management is a crucial area of interest with significant recent developments.

4. The review covers a wide range of neural interfaces, but it doesn't go into much detail about the specific challenges of each application. Including case studies or specific examples where 2D nanomaterials have been successfully used in particular applications, like deep brain stimulation or cortical recording.

We acknowledge the importance of the examples highlighted. While we have extensively discussed these instances throughout the manuscript, they are not concentrated or elaborated upon at length. Given the scope of this review, which focuses on the broad applications of 2D materials in neural interfaces (chemical, optical, and electrical), we believe that dedicating extensive text to specific applications may unbalance the article's structure.

5. The section on future research directions is a bit general and could be more specific. It would be helpful to outline concrete research questions and detailed experimental setups to address current gaps in knowledge. 

In section 4, we have supplemented the discussion by introducing the term "nanoarchitectonics" to describe the importance of designing nanomaterial structures for future interface design. This concept serves as a potential example and a direction for future research to fill knowledge gaps in interface materials. Such structured interface generation or modification methods may significantly improve interface properties.

Reviewer 2 Report

Comments and Suggestions for Authors

This is a good review article on interesting subjects. Publication of this review with this topic in IJMS is a good idea. I basically recommend publication of this review in IJMS. Some revisions are necessary for further improvements such as enhanced impacts and authors’ opinions. Please see below.

1) Although the contents of this review article are nice, initial impression and impact from the title are not so strong. Some spice words are necessary for the title. Inclusion of new concept words in the title would often give innovative impacts. I mat suggest use of an emerging conceptual term, nanoarchitectonics, in the title (for the concept, see https://academic.oup.com/bcsj/article/97/1/uoad001/7457599). For example, the title like ... 2D materials in neural interface: nanoarchitectonics and applications ... may sound more novel and innovative.

2) Figures are mostly made with assembly of the figures in the previous publications. Addition of more original figures may give better impression. From this viewpoint, Figure 1 is a nice figure. One more initial figure to explain contents and flow of this review article had better be added.

3) Similarly, one final figure to summarize conclusion part is asl recommended. Conclusion part is well written, Addition of one original figure for the authors’ opinions will give nice impacts.

Author Response

We sincerely appreciate the valuable feedback provided by the reviewer. We found these suggestions to be very helpful and have carefully addressed each point. Here are our responses:

1) Although the contents of this review article are nice, initial impression and impact from the title are not so strong. Some spice words are necessary for the title. Inclusion of new concept words in the title would often give innovative impacts. I mat suggest use of an emerging conceptual term, nanoarchitectonics, in the title (for the concept, see https://academic.oup.com/bcsj/article/97/1/uoad001/7457599). For example, the title like ... 2D materials in neural interface: nanoarchitectonics and applications ... may sound more novel and innovative.

Response 1: Thank you for your suggestion regarding the title. We agree that incorporating a term like "nanoarchitectonics" could be beneficial. However, since our article encompasses a broad range of nanomaterial concepts without focusing intensively on any specific new concept, we found it challenging to incorporate "nanoarchitectonics" or similar terms directly into the title. Nevertheless, we acknowledge the value of this suggestion and have included a discussion on the guiding significance of the term "nanoarchitectonics" in the conclusion section.

2) Figures are mostly made with assembly of the figures in the previous publications. Addition of more original figures may give better impression. From this viewpoint, Figure 1 is a nice figure. One more initial figure to explain contents and flow of this review article had better be added.

Response 2: We appreciate your recommendation to use original figures to describe the content and flow of the article. While we agree that this approach could be beneficial, we have focused on using Fig1 to organize and convey the key information and structure of the paper. We believe that the figures referenced in the article best illustrate the various advancements without additional comprehensive illustrations. Therefore, we have not added extra figures at this time.

3) Similarly, one final figure to summarize conclusion part is asl recommended. Conclusion part is well written, Addition of one original figure for the authors’ opinions will give nice impacts.

Response 3: We appreciate the suggestion to include an image in the conclusion section as a summary tool. However, given that Fig1 already serves as a comprehensive overview and outlines various aspects of interfaces that may generate potential future directions, we feel that creating another figure based on the conclusions may result in significant overlap with Fig1. Thus, we have decided not to add an additional image for the time being.